# Cross-Border Accessibility and Spatial Effects of China-Mongolia-Russia Economic Corridor under the Background of High-Speed Rail Environment

**DOI:** 10.3390/ijerph191610266

**Published:** 2022-08-18

**Authors:** Nanchen Chu, Xiangli Wu, Pingyu Zhang

**Affiliations:** 1College of Geographical Sciences, Harbin Normal University, Harbin 150025, China; 2Northeast Institute of Geography and Agroecology, Chinese Academy Sciences, Changchun 130102, China; 3College of Resources and Environment, University of Chinese Academy of Sciences, Beijing 100049, China

**Keywords:** accessibility, HSR environment, spatial pattern, China-Mongolia-Russia economic corridor

## Abstract

Under the background of “the Belt and Road” and “China-Mongolia-Russia economic corridor” initiatives, we studied the urban accessibility level and regional spatial effect of the west line and east line of China-Mongolia-Russia economic corridor in the high-speed rail (HSR) environment. The results are as following. (1) The operation of China-Mongolia-Russia HSR will greatly improve the urban accessibility level, which will shorten the whole journey time to two days along China-Mongolia-Russia economic corridor. The regional space-time convergence effect will be very strong in the China-Mongolia-Russia HSR environment. (2) The accessibility level and its improvement degree of the China-Mongolia-Russia east line are stronger than those of the west line. The accessibility level of different countries differs: China > Russia > Mongolia. The accessibility improvement degree of different countries also differs: Mongolia > Russia > China. Spatially, the accessibility improvement degree of the cities, which are located in the middle of the line is stronger than those cities at the beginning and end of the line. (3) Affected by the China-Mongolia-Russia HSR environment, the spatial polarization effect of China-Mongolia-Russia HSR axial belt will be further enhanced. The internal boundary effect of the China-Mongolia-Russia HSR axial belt will disappear. New HSR economic growth poles will occur, promoting the formation of point-axis system. China-Mongolia-Russia cross-border trade creation and transfer effects will be deepened.

## 1. Introduction

The high-speed rail (HSR) refers to the railway system with the operating speed of more than 200 km/h through the transformation of the original line, or the newly-built railway system with the operating speed of more than 250 km/h. HSR is a fast, safe, environmental friendly, high efficient, high aggregation, high technology, wide radiation and long-span modern transportation mode. The HSR operation marks that the railway transportation has entered a new modern era. HSR has become the important mode for people traveling with the advantages of high speed, large capacity, low energy consumption, comfort, safety, and great market competitiveness. The HSR opening has a profound impact on the spatial organization of cities and regions along the HSR line. The high space-time compression effect has brought the spatial transformation of urban and regional form, function, development mode. The HSR opening has also brought the gradual change and reconstruction of spatial pattern along the HSR line. In 2013, “the Belt and Road” initiative, including “the Silk Road Economic Belt” initiative and “the 21st-Century Maritime Silk Road” initiative were proposed by China. In “the Belt and Road” initiative, China-Mongolia-Russia economic corridor was one of the six major important economic corridors. In 2014, the construction of China-Mongolia-Russia economic corridor was proposed by the heads of China, Mongolia and Russia in Dushanb. In 2016, the planning outline for the construction of China-Mongolia-Russia economic corridor was signed by the heads of China, Mongolia and Russia officially. In 2019, the bilateral relation of China and Russia was upgraded to the “comprehensive strategic coordination partnership in the new era”. Among the Northeast Asian countries, China-Mongolia-Russia economic corridor is a key specific administrative corridor area. There are two channels along the China-Mongolia-Russia economic corridor. (1) The China Northeast channel, from Beijing, Tianjin, Hebei and Inner Mongolia in China, via Erenhot to Zamyn-Uud, through Mongolia to Russia. (2) The China North channel, from Northeast China to Manzhouli, via Zabaykalsk to Russia. Restricted by the backward cross-border infrastructure, serious traffic channel bottlenecks, and lack of HSR modern equipment, there is strong demand for realizing the HSR interconnection along the two channels of China-Mongolia-Russia economic corridor.

As for China, China’s HSR construction was begun with the promulgation of medium and long term railway network planning in 2004. China’s first HSR line (Beijing-Tianjin intercity HSR line) was operated in 2008. China’s “eight vertical and eight horizontal HSR network” blueprint was implemented in 2016. China’s HSR operation mileage was reached 29,000 km, exceeding 2/3 of the world’s total HSR mileage, with the longest HSR mileage in the world at the end of 2018. At present, China’s HSR network has covered most metropolitan areas with the population of more than 2 million [1]. The scope of one-day HSR communication circle has been expanded significantly in central cities [2], forming a “multi-center pattern” led by HSR national service centers such as Shanghai, Beijing and Guangzhou, and an increasingly large and complex spatial cluster with HSR characteristics [3]. Yangtze River Delta, Pearl River Delta, and Beijing-Tianjin-Hebei urban agglomeration have become the significant and important HSR hub areas [4]. As for Russia, Russia’s railway transportation development has a long history. The first passenger railway line was operated (St. Petersburg-Tsarskoe Selo) in 1837. The Siberian Railway Line was begun to construct in 1891 [5]. The railway Order No. 384 of 10-year three-stage railway reform plan was issued in 2001 [6]. The Russia’s Railway Corporation was established in 2003. The outline for rapid and HSR network development before 2020 was formulated in 2006. The railway development strategy for 2008–2030 was implemented in 2007. The new railway lines of 20,500 km (including 1528 km HSR lines) were planned to construct [7]. The first “Peregrine Falcon” (Сапсан) HSR No.1 was operated (daily 7 HSR pairs from Moscow to St. Petersburg and daily 2 HSR pairs from Moscow to Nizhny Novgorod) in 2009 [8]. The daily 2 HSR pairs from St. Petersburg to Helsinki was operated at the end of 2010. At present, Russia has a railway mileage of 86,000 km [9] and a railway network density of 0.005 km/km^2^. Russian railways account for 80% of the national freight turnover and 70% of the national passenger turnover [6]. As for Mongolia, compared with China and Russia, there has been a relatively slow railway development in Mongolia. The first railway line (Ulaanbaatar-Nalaikha coal mine) was opened in 1937. The Ulaanbaatar railway line was begun to construct in 1947. The Ulaanbaatar-Naushki railway line was operated in 1949. The Ulaanbaatar-Zamyn Uud railway was begun to get into operation in 1955 [10,11]. The railway privatization reform dominated by state ownership was carried out in 1994. The long-term plan for the national railway network expansion, which planned to construct four railway lines in the eastern Mongolia, western Mongolia, Mongolian Gobi, traverse eastern and western Mongolia, was formulated in 2008. The national railway transportation development plan was promulgated in 2010 [12]. At present, Mongolia has a railway mileage of 2500 km and a railway network density of 0.16 km/100 km^2^ [13]. The two trunk railway lines (from Ulaanbaatar to Mongolia-Russian border port, from Choybalsan to Mongolia-Russian border port), and three branch railway lines have been into operation in Mongolia [14]. Mongolia’s railways account for 3/4 of the national freight volume and 1/3 of the national passenger volume [10,15]. Mongolia has a railway freight volume of 25.763 million tons, a railway passenger volume of 2.6 million people, and a railway operation revenue of 616 billion tugrik.

As an important corridor area in Northeast Asia, the China-Mongolia-Russia economic corridor has highly complementary resource advantages, great production capacity cooperation potential, and wide international investment cooperation space [16]. At present, the researches on China-Mongolia-Russia economic corridor focus on the economic and trade development [17,18,19], the energy and mineral cooperation [20,21,22,23], the cross-border logistics and transportation [24], the tourism resource sharing [25], the infrastructure construction [26,27], and the cultural industry identification [28]. Wen discussed the trade development history, the import and export products pattern, and the trilateral trade potential of China, Mongolia and Russia [17]. Violin proposed potential joint infrastructure projects of manufacturing, technology and service industries under the framework of China-Mongolia-Russia economic corridor [18]. Fartyshev proposed to establish a cross-border Russian-Mongolian production complex on the basis of developing efficient production facilities in Siberia and Mongolia [19]. Yang studied the basis, opportunities, situation and challenges of China-Mongolia-Russia energy cooperation from the perspectives of supply demand relations, energy policy, labor division, regional development policy and international geo-environment [20]. Wei, Khomich, and Abalakov explored the development utilization status, industrial structure upgrading and future cooperation potential of mineral resources such as steel, cement and uranium metallogenic belt [21,22,23]. Zheng analyzed the cross-border logistics transportation problems, exploring the cooperation mechanism to promote the cross-border logistics transportation facilitation along China-Mongolia-Russia economic corridor [24]. Zhou studied the spatial pattern and influencing factors of natural tourism resources, exploring the homogeneity and complementarity of natural tourism resources along China-Mongolia-Russia economic corridor [25]. Dong scientifically designed the China-Mongolia-Russia HSR line, proposing a mutually beneficial and win-win innovative cooperation mode through “HSR for resources and HSR for market” [26,27]. Sun believed that China, Mongolia and Russia should follow the principle of multicultural identity, implementing the planning guidance, culture first, highlight characteristics, building brands, strengthening cooperation in talent training, scientific research, international platform construction [28].

The HSR construction has the important significance to improve the backward cross-border infrastructure and transportation bottlenecks along the China-Mongolia-Russia economic corridor. As for China, Mongolia and Russia, HSR construction could deepen the cooperation of employment, trade, industries, science, technology, logistics, tourism, and talent training. It also could expand the passenger flow and the urban scale along the China-Mongolia-Russia economic corridor. China-Mongolia-Russia HSR construction is an important meeting point for the implementation of China’s Northeast Revitalization Strategy, Mongolia’s Grassland Silk Road, Russia’s Eurasian Economic Union, Russia’s Far East Development Strategy and Tea Ceremony Revitalization Plan. As for China, China-Mongolia-Russia HSR construction could strengthen the resources and energy security. It could open up the overcapacity export market of China’s northeast old industrial base, accelerating the transformation of resource-based cities in Northeast China. As for Mongolia, China-Mongolia-Russia HSR construction could construct a new opening-up channel. It could break the inland closure, strengthening the connection with developed regions such as East Asia and Europe. As for Russia, China-Mongolia-Russia HSR construction could meet with Russia’s “strategic eastward move” measures, accelerating the economic integration process in the Asia Pacific region. HSR infrastructure connectivity is a priority area for the construction of China-Mongolia-Russia economic corridor. So it is urgent to study the layout and construction of China-Mongolia-Russia HSR lines. In this paper, under the background of “the Belt and Road” and “China-Mongolia-Russia economic corridor” initiatives, we studied the urban accessibility level and regional spatial effect of the west and east lines along China-Mongolia-Russia economic corridor in the HSR environment. First, the urban accessibility level and its dynamic changes with and without China-Mongolia-Russia HSR environment were evaluated by using two indicators, i.e., the weighted average time, and the economic potential. Then after the evaluation, the regional differentiation pattern of the accessibility level and its dynamic changes with and without China-Mongolia-Russia HSR environment were simulated respectively by using ArcGIS. Finally, affected by the China-Mongolia-Russia HSR accessibility, the regional spatial effect were analyzed from the perspectives of polarization effect, boundary effect, integration effect, trade creation and transfer effect, etc. In the theory, this study creates a theoretical analysis framework of HSR accessibility and its spatial effect in the cross-border “corridor” zone with the geographical characteristics. And then it applies the China-Mongolia-Russia HSR accessibility to the practice of cross-border regional research along the China-Mongolia-Russia economic corridor. This study could enrich the theoretical systems of geo-economics, transportation geography, and urban geography. At the same time, it also could provide theoretical reference and guidance for other cross-border regional research in the world, especially in the other economic corridors of “the Belt and Road” initiative. In the practice, on the basis of studying the China-Mongolia-Russia HSR accessibility and analyzing the spatial effects such as space-time reduction, location enhancement, border breakthrough, core-edge, trade creation and transfer along the China-Mongolia-Russia economic corridor, we propose the HSR spatial development mode of cross-border cooperation in the China-Mongolia-Russia transport economic corridor, which provides a scientific basis for bilateral or multilateral regional cooperation among China, Mongolia and Russia. In addition, this study could also provide scientific reference for regional development planning, economic optimization layout, energy and resource development under the background of HSR construction along China-Mongolia-Russia economic corridor. It also could provide policy implications for the border trade, transportation facilities, border tourism, border cooperation zone, eco-environment protection along China-Mongolia-Russia economic corridor.

## 2. Materials and Methods

### 2.1. Study Area

According to the regional scope of China-Mongolia-Russia international economic corridor [25], the western and eastern lines of China-Mongolia-Russia HSR [26,27], the channel paths of China-Mongolia-Russia economic corridor [29], the research area and research route is shown in Figure 1. There are two channels along the China-Mongolia-Russia economic corridor. (1) China-Mongolia-Russia west line: Shijiazhuang-Tianjin-Beijing-Ulanqab-Erenhot-Zamyn Uud-Sainshand-Choyr-Ulaanbaatar-Darhan-Sukhbaatar. (2) China-Mongolia-Russia east line: Dalian-Shenyang-Changchun-Harbin-Qiqihar-Manzhouli-Zabaykalsk-Chita. China-Mongolia-Russia west and east lines gather in Ulan Ude, via Irkutsk, Krasnoyarsk, Novosibirsk, Omsk, Tyumen, Yekaterinburg, Perm, Kirov, Nizhny Novgorod, Vladimir, Moscow, and finally arriving in St. Petersburg. The China-Mongolia-Russia west and east lines cross 30 provincial units (Russia 18 + China 7 + Mongolia 5), and 39 urban units such as Beijing, Ulaanbaatar and Moscow. At present, Beijing-Shijiazhuang HSR section and Qiqihar-Harbin-Dalian HSR section have been into operation. In the future, with the opening of China-Mongolia-Russia HSR west line, Beijing-Tianjin-Hebei Urban Agglomeration and Bohai Economic Rim will export the industrial and technical elements to Mongolia and Russia conveniently. The resource, energy and international production capacity cooperation markets of Russia and Mongolia will get great development. With the operation of China-Mongolia-Russia HSR east line, a new economic engine for bilateral regional development cooperation will be created between China and Russia, meeting with the China’s “Northeast Revitalization initiative” and Russia’s “Far East Development initiative”.

### 2.2. Research Methods

The “accessibility” concept was proposed by Hansen for the first time, which was the interaction opportunity among cities in the transportation network [30]. Accessibility also refers to the ability of arriving at the appointed place in the appropriate time by some means of transportation [31]. It also means the people’s mobility and the mobile opportunities to reach the destination. Accessibility contains the temporal meaning, spatial concept, economic value, social value, starting point, terminal point, transportation system and other factors [32]. In this paper, the accessibility refers to the external accessibility of residents from the starting city station to the terminal city station by taking common speed train (CST) or HSR in the specific time. It does not consider the internal accessibility of residents from the starting place to the station, and from the station to the destination place through bus, subway, light rail and other transportation modes [33]. At the same time, the urban CST and HSR accessibility are studied along the China-Mongolia-Russia west and east lines, rather than the urban comprehensive and complete accessibility.

#### 2.2.1. Railway Accessibility Indicators

“Accessibility” methods can be summarized as the travel cost and the location attraction. Based on the research objects and research level, the weighted average travel time, and the economic potential are selected to analyze the urban railway accessibility changes along the China-Mongolia-Russia west and east lines in the HSR environment. The weighted average travel time and the economic potential are the most commonly used indicators to measure the accessibility level and its dynamic changes. The regional differentiation pattern of the weighted average time, the economic potential and their dynamic changes are simulated respectively by using ArcGIS. At the same time, China-Mongolia-Russia HSR west and east lines have the starting point, the terminal point, the means of transportation and other factors, meeting with the basic accessibility conditions proposed by Li [32].

#### 2.2.2. The Weighted Average Travel Time

The weighted average travel time characterizes the physical sense of accessibility, which refers to a kind of time accessibility measurement from fixed city to other cities. It focuses on measuring the time accessibility level from the perspectives of space-time distance and cost saving, which visually represents the urban accessibility improvement degree. It’s intuitive, simple and commonly used to represent the accessibility improvement degree. It’s associated with several major factors of regions and cities along the HSR lines. These factors include (1) regional and urban transportation infrastructure improving degree, (2) regional and urban economic development and comprehensive competitive strength, (3) regional and urban scale, (4) regional and urban spatial location, and (5) transportation modes from fixed city to other cities. In general, the weighted average travel time of the regional economic centers is lower than that of the marginal areas. The weighted average travel time is higher, the time accessibility is lower, vice versa. The formula is as follows:(1)$$A_{i}=\frac{\sum_{j=1}^{n}\left(T_{ij}*M_{j}\right)}{\sum_{j=1}^{n}M_{j}}$$
where *A_i_* is the weighted average travel time of *i* city. *T_ij_* is the travel time between *i* city and *j* city by CST or HSR, *T_ij_* = *T_tij_* + *T_wi_*, *T_tij_* is the operation time of CST or HSR from *i* city station to *j* city station, and *T_wi_* is the retention time of CST or HSR at *i* city station [33]. *M_j_* is the population flow and economic flow of *j* city, which can reflect the radiometric force of *j* city to surrounding cities. *M_j_* is measured by GDP and population. $M_{j}=\sqrt{POP_{j}\times ECO_{j}}$, *POP_j_* is the population number of *j* city, *ECO_j_* is the year-end deposit balance of residents of *j* city (Mongolia’s monetary unit is tugrik, and Russia’s monetary unit is ruble. Mongolia’s tugrik and Russia’s ruble are converted into China’s RMB). *ECO_j_* can reflect the purchasing power of *j* urban residents, which could represent the financial resources and willingness to choose HSR for China-Mongolia-Russia cross-border traveling. Due to the data restrictions, GDP and other economic indicators are not selected. *n* is the total number of the cities except *i* city.

#### 2.2.3. The Economic Potential

Economic potential characterizes the economic sense of accessibility, which refers to a kind of economic accessibility measurement from fixed city to other cities. It analyzes the inter-regional interaction and distance attenuation due to the gravity effect from the economic and social perspectives, which reflects the radiation force of the fixed city to other cities along the transportation line. Economic potential makes up the mutual separation conditions of accessibility, which is estimated by the weighted average travel time. It is a comprehensive indicator of economic strength, development potential, connection degree with the external environment, and HSR situations. On one hand, urban accessibility is calculated based on economic potential. On the other hand, the agglomeration or diffusion trend of flow is shown by economic potential. So economic potential is explained as the total economic activity caused by the opening of HSRs in a given area and specific time. It’s associated with several major factors of regions and cities along the HSR lines. These factors include (1) regional and urban spatial location, (2) regional and urban economic quality and economic strength, (3) economic cost, distance, and time from fixed city to the economic center, and (4) regional and urban economic radiometric force, and economic scale. The economic potential is higher, the economic accessibility is higher, vice versa. The formula is as follows:(2)$$P_{i}=\sum_{j=1}^{n}\frac{M_{j}}{T_{ij}{}^{a}}$$
where *P_i_* is the economic potential of *i* city. *a* is the distance friction coefficient or elastic attenuation coefficient. *a* is 1, referring to the urban interaction in the national scale. *a* is 2, referring to the urban interaction in the regional scale [34,35]. The study area contains China, Mongolia and Russia, and *a* is taken as 1. *M_j_*, *T_ij_*, and *n* ditto.

### 2.3. Data Sources

The data sources are shown in Table 1.

## 3. Results of Accessibility

The China-Mongolia-Russia west line is 9044 km, with the average operating speed of 62.5 km/h and the travel time of 144.7 h. After the operation of China-Mongolia-Russia HSR west line, the travel time will be reduced from 144.7 h to 45.2 h, with the time distance decreasing by 68.75%. The China-Mongolia-Russia east line is 9139 km, with the average operating speed of 61.4 km/h and the travel time of 148.8 h. After the operation of China-Mongolia-Russia HSR east line, the travel time will be reduced from 148.8 h to 44.7 h, with the time distance decreasing by 69.98%. The China-Mongolia-Russia HSR opening has greatly reduced the intercity travel time along the China-Mongolia-Russia economic corridor. With strong temporal and spatial convergence effect, and remarkable accessibility improvement effect, the 2 days “HSR communication circle” will be distributed along the China-Mongolia-Russia economic corridor,.

After the China-Mongolia-Russia HSR opening, the travel time will be reduced from 129.7 h to 39.3 h between Beijing and Moscow. And the 2 days “HSR communication circle” will be distributed along Beijing-Moscow travel journey. The travel time will be reduced from 98.6 h to 31.5 h between Ulaanbaatar and Moscow. And the 2 days “HSR communication circle” will be distributed along Ulaanbaatar-Moscow travel journey. The travel time will be reduced from 31.1 h to 7.8 h between Beijing and Ulaanbaatar. And the 8 h “HSR communication circle” will be distributed along Beijing-Ulaanbaatar travel journey. What’s more, the 1 h “HSR communication circle” will be distributed among the most adjacent cities (Moscow-Vladimir, Sukhbaatar-Darhan, Zamyn Uud-Erenhot, Zhangjiakou-Beijing, Beijing-Tianjin, Zabaykalsk-Manzhouli, Manzhouli-Hulunbuir, Qiqihar-Daqing, Daqing-Harbin, Harbin-Changchun, Tieling-Shenyang).

### 3.1. Weighted Average Travel Time and Economic Potential Analysis

According to Table 2 and Table 3, after the China-Mongolia-Russia HSR opening, urban accessibility will be improved along the China-Mongolia-Russia economic corridor. The urban accessibility level and urban accessibility improvement degree of the China-Mongolia-Russia east line are slightly stronger than those of the China-Mongolia-Russia west line. (1) Along the China-Mongolia-Russia west line, the CST accessibility and HSR accessibility of China are stronger than those of Russia and Mongolia. The weighted average travel time of different countries differs: China < Mongolia < Russia. The economic potential of different countries also differs: China > Russia > Mongolia. However, after the opening of China-Mongolia-Russia HSR west line, the accessibility improvement degree of Mongolia is stronger than that of China and Russia. The weighted average travel time decreasing rate of different countries differs: Mongolia > Russia > China. The economic potential increasing rate of different countries also differs: Mongolia > China > Russia. The weighted average travel time of all the cities will be decreased from 63.1 h to 18.6 h, with the average decreasing rate of 70.6%. The economic potential of all the cities will be increased from 1046 to 2980, with the average increasing rate of 298%. (2) Along the China-Mongolia-Russia east line, the CST accessibility and HSR accessibility of China are stronger than those of Russia. The weighted average travel time of different countries differs: China < Russia. The economic potential of different countries also differs: China > Russia. However, after the opening of the China-Mongolia-Russia HSR east line, the accessibility improvement degree of Russia is stronger than that of China. The weighted average travel time decreasing rate of different countries differs: Russia > China. The economic potential increasing rate of different countries also differs: Russia > China. The weighted average travel time of all the cities will be decreased from 60.1 h to 17.8 h, with the average decreasing rate of 70.3%. The economic potential of all the cities will be increased from 1707 to 4189, with the average increasing rate of 351%. (3) As for Mongolia, compared with China and Russia, Mongolia has the shortest railway distance. However, Mongolia’s cities have the imperfect railway transportation system, aging railway equipment, low railway electrification level, under maintained railway roadbed and track. The Mongolia’s railway carrying capacity is far from meeting its railway transportation requirements. Therefore, the China-Mongolia-Russia HSR opening will contribute to the significant accessibility improvement in Mongolia’s cities. As for China, HSRs have been put into operation in the Beijing-Shijiazhuang section and Qiqihar-Harbin-Dalian section. China’s cities have perfect HSR technology and integration capacity. The HSR maximum speed of China’s cities can reach 350 km/h. As for Russia, HSRs have been put into operation only in the St. Petersburg-Moscow-Nizhny Novgorod section. The HSR speed of St. Petersburg-Moscow-Nizhny Novgorod section could reach only 200 km/h. Therefore, the accessibility improvement of Russia will be stronger than that of China in the future.

### 3.2. Spatial Pattern Analysis of Accessibility

According to Figure 2, Figure 3, Figure 4 and Figure 5, before and after the opening of the China-Mongolia-Russia HSR, the superior areas of CST accessibility and HSR accessibility show a diffusion trend along the China-Mongolia-Russia west and east lines. The CST accessibility and HSR accessibility pattern characterized by the weighted average travel time and economic potential show the imbalance differentiation characteristics of “high beginning, high end, and low middle” along the China-Mongolia-Russia economic corridor. The CST accessibility and HSR accessibility of the cities, which are located in the middle of the lines is weaker than those cities at the beginning and end of the lines. The accessibility superior elements show a decreasing trend from Chinese cities, via the China-Mongolia border and China-Russia border, to Russia. However, with the superior economic growth potential, Moscow and St. Petersburg occupy the important economic developed positions along the China-Mongolia-Russia economic corridor. But the accessibility radiation and transmission effects of Moscow and St. Petersburg are far less than those of Chinese cities.

According to Figure 2, Figure 3, Figure 4 and Figure 5, before and after the opening of the China-Mongolia-Russia HSR, the accessibility improvement pattern characterized by the weighted average travel time decreasing rate and economic potential increasing rate show the imbalance differentiation characteristics of “high middle, low beginning, and low end” along the China-Mongolia-Russia economic corridor. The accessibility improvement of the cities, which are located in the middle of the lines is stronger than those cities at the beginning and end of the lines. The accessibility improvement superior areas are concentrated in the series areas of Ulan Ude-Irkutsk-Krasnoyarsk-Novosibirsk-Omsk, with their weighted average travel time decreasing rate more than the average of whole China-Mongolia-Russia economic corridor. Besides, the accessibility improvement superior areas are also distributed in the Erenhot-Zamyn Uud (border ports of China and Mongolia) and Manzhouli-Zabaykalsk (border ports of China and Russia), with their economic potential increasing rate more than 1000%.

At present, the 1435 mm standard rail is used in China. The 1524 mm wide rail is used in Mongolia and Russia. There are long time and high cost for rail track changing and docking among the China-Mongolia border cities and China-Russia border cities. With the China-Mongolia-Russia HSR opening, the automatic intelligent conversion and driving will be realized between standard rail and wide rail. The substantial reduction of time distance will promote the substantial accessibility growth among the China-Mongolia border cities and China-Russia border cities. In the future, with the infrastructure interconnection between China-Mongolia-Russia economic corridor and Northeast Asia, China’s initiative of opening up in the border, Mongolia’s initiative of rejuvenating the country through mining, Russia’s initiative of Far East development all will be greatly promoted. The China-Mongolia-Russia HSR will attract the cross-border flow of talents, economy, science and technology, accelerating the industrial layout optimization and the economic trade cooperation upgrading along the China-Mongolia-Russia economic corridor. It will also excavate the urban economic potential and urban core competitiveness, creating the peaceful, stable, equal, trusting, beneficial, opening, win-win-win geo-strategic pattern in Northeast Asia under the background of economic globalization.

## 4. Spatial Effects Based on HSR Accessibility

### 4.1. Spatial Polarization Effect of China-Mongolia-Russia HSR Axial Belt Will Be Further Enhanced

After the China-Mongolia-Russia HSR opening, the weighted average travel time decreasing rate of China-Mongolia-Russia HSR west line will reach 70.6%. The weighted average travel time decreasing rate of China-Mongolia-Russia HSR east line will reach 70.3%. The economic potential increasing rate of China-Mongolia-Russia HSR west line will reach 298%. The economic potential increasing rate of China-Mongolia-Russia HSR east line will reach 351%. Urban accessibility and regional accessibility will be improved along the China-Mongolia-Russia economic corridor. The strong space-time compression effect brought by the China-Mongolia-Russia HSR opening will be strengthened, attracting the population, economy, industry, technology and other factors of non-HSR axial belt gathering along the HSR axial belt. The spatial polarization effect of China-Mongolia-Russia HSR will be further enhanced affected by the significant siphon effect. In 2018, Mongolia’s provinces which were distributed along the China-Mongolia-Russia economic corridor, had 25.1% of the Mongolia’s territory, 62.2% of the Mongolia’s population and 58.2% of the Mongolia’s labor force, creating 75.3% of Mongolia’s GDP, 59.5% of Mongolia’s local government revenue, 60.4% of Mongolia’s local government expenditure, 68.1% of Mongolia’s industrial sales, 87.7% of Mongolia’s deposits and 92.6% of Mongolia’s retail trade volume. And their proportion of population, labor force, GDP, local government income, local government expenditure, retail trade volume and industrial sales in Mongolia had increased by 0.67%, 5.32%, 1.44%, 0.81%, 2.17%, 2.02% and 6.76% respectively compared with 2010. In 2018, Russia’s federal subjects which were distributed along the China-Mongolia-Russia economic corridor, had 41.8% of the Russia’s territory, 47.1% of the Russia’s population and 50.4% of the Russia’s employment, creating 63.0% of the Russia’s regional GDP, 60.6% of the Russia’s economic fixed assets, 63.3% of the Russia’s industrial output value, 70.4% of the Russia’s mining output value, 61.1% of the Russia’s manufacturing output value, 57.9% of the Russia’s of power and gas industry output value, 54.2% of the Russia’s retail trade volume and 58.6% of the Russia’s fixed capital investment. And their proportion of population, employment, industrial output value, mining output value, manufacturing output value, power and gas industry output value and fixed capital investment had increased by 0.30%, 1.90%, 2.17%, 0.90%, 1.83%, 2.31% and 8.96% respectively compared with 2010. The spatial agglomeration of these population and economic factors will promote the “core-edge” pattern of HSR core axial belt areas and non-HSR edge areas along the China-Mongolia-Russia economic corridor. The population and economic development of HSR axial belt areas will be faster than that of non-HSR axial belt areas. The spatial agglomeration of the population and economic factors will be very slow in the non-HSR axial belt areas, especially in Mongolia. The population and economic gravity centers will move to the China-Mongolia-Russia HSR axial belt. The spatial structure will become more and more unbalanced along the China-Mongolia-Russia corridor.

### 4.2. Internal Boundary Effect of the China-Mongolia-Russia HSR Axial Belt Will Disappear

There has been the long-term close communication in history, culture, economy and society among China, Mongolia and Russia. China, Mongolia and Russia have significant geographical advantages, good political relations and fine economic trade cooperation. As for China and Mongolia, the trade volume had increased from 3.96 billion dollars in 2010 to 7.99 billion dollars in 2018. As for China and Russia, the trade volume had increased from 55.53 billion dollars in 2010 to 107.06 billion dollars in 2018. China has been the largest trading partner and investor of Mongolia. China has also been the largest trading partner of Russia. These phenomena will provide the important foundation for the construction of China-Mongolia-Russia HSR west and east lines. After the China-Mongolia-Russia HSR west line opening, the weighted average travel time decreasing rate will be 69.3%, 71.4% and 70.8% respectively in China, Mongolia and Russia. The economic potential increasing rate will be 355%, 430% and 210% respectively in China, Mongolia and Russia. After the China-Mongolia-Russia HSR east line opening, the weighted average travel time decreasing rate will be 69.5% and 70.9% respectively in China and Russia. The economic potential increasing rate will be 276% and 410% respectively in China and Russia. The space-time distance reduction will weaken the communication barriers in the border cities of China, Mongolia and Russia. Various of population and economic factors will break the traditional administrative boundaries among China, Mongolia and Russia. Industry, employment, logistics, capital, technology, transportation, information and other factors will be flowed in the cross-border space among China, Mongolia and Russia. The internal boundary effect of China-Mongolia-Russia HSR axial belt will disappear. For example, in 2018, the cargo volume of Manzhouli port (border port of China and Russia) was 31.92 million tons, with the increasing rate of 3% compared with the last year. The total import and export value of Manzhouli port was 35.4 billion RMB, with the increasing rate of 6.7% compared with the last year. The number of inbound and outbound people of Manzhouli port was 1.92 million, with the increasing rate of 3.2% compared with the last year. In 2018, the import and export goods of Erenhot port (border port of China and Mongolia) was 12.03 million tons, with the increasing rate of 17.16% compared with the last year. The total foreign trade value of Erenhot port was 23.80 billion RMB, with the increasing rate of 14.9% compared with the last year. The import and export value of Erenhot to Mongolia reached 16.48 billion RMB, with the increasing rate of 17.9% compared with the last year. The import and export value of Erenhot to Russia reached 7.28 billion RMB, with the increasing rate of 8.4% compared with the last year. After the China-Mongolia-Russia HSR opening, the China-Mongolia-Russia economic corridor will have the coordinated and unified systems, cultures, policies and laws. China, Mongolia and Russia will have more deepening cooperation in labor resources, industrial layout, science and technology, transportation infrastructure, etc. The internal boundary effect of China-Mongolia-Russia HSR axial belt will disappear, and the integration effect of boundary areas and boundary cities will appear along the China-Mongolia-Russia economic corridor.

### 4.3. New HSR Economic Growth Poles Will Occur, Promoting the Formation of Point-Axis System

Beijing, Moscow, Tianjin, Shijiazhuang, Harbin, Changchun, Shenyang, Dalian, and St. Petersburg have 65.7% of the population and 83.7% of the residents’ deposit balance along the China-Mongolia-Russia economic corridor. Compared with other cities, these cities have the highest population and economic development along the China-Mongolia-Russia economic corridor. Before and after China-Mongolia-Russia HSR west and east lines opening, Beijing, Tianjin, Shijiazhuang, Harbin and Changchun have the shortest weighted average travel time. Beijing, Tianjin, Shijiazhuang, Moscow, St. Petersburg, Shenyang, Harbin, Changchun and Dalian have the greatest economic potential. With superior economic conditions, mature industrial clusters, developed transportation system and perfect social facilities, these cities have become the economic growth poles along the China-Mongolia-Russia economic corridor. China-Mongolia-Russia HSR east-west intersection cities (Ulan Ude, Irkutsk), China-Mongolia-Russia border port cities (Manzhouli, Zabaykalsk, Erenhot, Zamyn Uud), China-Mongolia-Russia resource and energy rich cities (Krasnoyarsk, Mongolia’s cities) will have the highest weighted average travel time decreasing rate and the highest economic potential increasing rate. These cities will have significantly improved accessibility, gradually developing into the new economic growth poles along the China-Mongolia-Russia economic corridor. The new and old economic growth poles will promote the “multi-core” development pattern along the China-Mongolia-Russia economic corridor. Population, economy, technology and other factors of the “multi-core” will spillover and diffusion, promoting the formation of point-axis system along the China-Mongolia-Russia economic corridor. However, the marginal areas between the “multi-core” and the marginal areas outside both have weak accessibility level. Different areas will have the different accessibility potentials. Various of economic activities will be carried out in the different imbalance accessibility potential areas, forming the multiple “core-edge” patterns within the point-axis system along the China-Mongolia-Russia economic corridor.

### 4.4. China-Mongolia-Russia Cross-Border Trade Creation and Transfer Effects Will Be Deepened

With the opening of China-Mongolia-Russia HSR, the bilateral and multilateral division of labor and cooperation will get continuous deepening along the China-Mongolia-Russia economic corridor. At the same time, trade cooperation will become more and more frequent among China, Mongolia, Russia and Northeast Asian countries, producing the strong trade creation effect. In 2017, compared with the last year, as for China and Japan, the import and export volume of goods reached 303.05 billion dollars, with the increasing rate of 10.17%. As for China and South Korea, the import and export volume of goods reached 280.26 billion dollars, with the increasing rate of 10.90%. As for China and Russia, the import and export volume of goods reached 84.22 billion dollars, with the increasing rate of 20.98%. As for China and Mongolia, the import and export volume of goods reached 6.40 billion dollars, with the increasing rate of 38.85%. As for Mongolia and Russia, the import and export volume of goods reached 1.29 billion dollars, with the increasing rate of 37.25%. As for Mongolia and Japan, the import and export volume of goods reached 378 million dollars, with the increasing rate of 9.67%. As for Mongolia and South Korea, the import and export volume of goods reached 209 million dollars, with the increasing rate of 1.44%. With the border effect disappearance and the the integration effect diffusion in the border areas, the regional transaction costs will decrease among China, Mongolia and Russia. The regional trade cooperation of the China-Mongolia-Russia economic corridor and other countries will be partially transferred to the interior of the China-Mongolia-Russia economic corridor, producing the strong trade transfer effect. The scale of cross-border investment will continue to grow among China, Mongolia and Russia. In 2017, compared with the last year, China invested 1.55 billion dollars in Russia, with the increasing rate of 19.72%, reaching the direct investment stock of 13.87 billion dollars. The turnover of China’s contracted projects to Russia was 1.99 billion dollars, with the increasing rate of 34.07%. China sent 2181 people to Russia for labor cooperation, with the increasing rate of 109.71%. In 2018, compared with the last year, China invested 437 million dollars in Mongolia, with the increasing rate of 33.64%, reaching the direct investment stock of 5.96 billion dollars. In 2017, compared with the last year, the turnover of China’s contracted projects to Mongolia was 1.07 billion dollars, with the increasing rate of 43.21%. China’s labor cooperation with Mongolia reached 822 people abroad, with the increasing rate of 38.62%. The investment scale will also expand between the internal China-Mongolia-Russia economic corridor and the external China-Mongolia-Russia economic corridor. In 2017, compared with the last year, China invested 444 million dollars in Japan, with the increasing rate of 29.08%. The turnover of China’s contracted projects to Japan was 328 million dollars, with the increasing rate of 4.89%. In 2018, compared with the last year, Japan invested 244 million dollars in Mongolia, with the increasing rate of 165.47%. South Korea invested 27 million dollars in Mongolia, with the increasing rate of 164.71%. All these phenomena will boosted the long-term economic and trade development between China-Mongolia-Russia economic corridor and Northeast Asian countries.

## 5. Discussion

Compared with earlier studies [26,27], Dong found that the Russia’s Zabaikalsk Territory, Russia’s Republic of Buryatia, Mongolia and other regions have backward economic development, slow economic growth, negative population growth, weak transportation infrastructure and lack of funds. Therefore, these areas have low economic accessibility. In this paper, compared with the Northeast China, Russia’s Moscow, Russia’s St. Petersburg and other developed areas, Russia’s Zabaikalsk Territory, Russia’s Republic of Buryatia, Mongolia, which are located in the middle of the China-Mongolia-Russia HSR line, have very weak economic accessibility indeed. This finding support the Dong’s conclusions. However, Dong’s research area only covers 8 research units of Beijing, Hebei, Inner Mongolia, Heilongjiang, Mongolia, Zabaikalsk Territory, Republic of Buryatia and Irkutsk Region. In this paper, we select 30 provincial units along the west and east lines of China-Mongolia-Russia economic corridor. So our research conclusions not only support Dong’s research conclusions, but also are more convincing and comprehensive to a certain extent. Compared with earlier study [33], Chu found that the railway accessibility of the nodes, which are located at the beginning and end of the railway line, is weaker than those nodes located in the middle of the line along the China-Mongolia-Russia HSR east line. In this paper, we also came to the conclusion that the CST accessibility and HSR accessibility pattern characterized by the weighted average travel time and economic potential show the unbalanced differentiation characteristics of “high beginning, high end, and low middle” along China-Mongolia-Russia economic corridor. This current study result supports the Chu’s finding. In the future, based on the China-Mongolia-Russia HSR opening, Northeast China will connect the Sino-Russian international transport corridors of “Primorsky No.1” and “Primorsky No.2”. Relying on the China’s Harbin important hub and Harbin-Mudanjiang HSR, China-Mongolia-Russia economic corridor will build the branch line of Harbin-Mudanjiang-Suifenhe-Dongning (starting from China’s Harbin, passing through Mudanjiang, Suifenhe and Dongning, radiating Russia’s Vladivostok, Vostochnyy and Japan’s Niigata). Relying on the China’s Harbin important hub and Harbin-Jiamusi HSR, China-Mongolia-Russia economic corridor will build the branch line of Harbin-Jiamusi-Shuangyashan-Tongjiang (starting from China’s Harbin, passing through Jiamusi, Shuangyashan, Hegang, connecting Russia’s Khabarovsk Territory and Jewish Autonomous Area via China’s Tongjiang, Fuyuan, Luobei, Raohe and other ports). Relying on the China’s Harbin important hub and Harbin-Heihe railway, China-Mongolia-Russia economic corridor will build the branch line of Harbin-Suihua-Bei’an-Heihe (starting from China’s Harbin, Suihua, Bei’an and Heihe, connecting Russia’s Blagoveshchensk through Heilongjiang Highway Bridge, radiating Russia’s Amur Region and Republic of Sakha(Yakutia)). Relying on the China’s Changchun important hub and Changchun-Hunchun HSR, China-Mongolia-Russia economic corridor will build the branch line of Changchun-Jilin-Tumen-Hunchun (starting from China’s Changchun, passing through Jilin, Tumen and Hunchun, radiating the Russian important ports such as Kraskino, Zarubino, Vladivostok in the Primorsky Territory). Relying on the China’s Shenyang important hub and Shenyang-Dandan HSR, China-Mongolia-Russia economic corridor will build the branch line of Shenyang-Benxi-Dandong-Sinuiju (starting from China’s Shenyang, passing through Benxi, Dandong, North Korea’s Sinuiju, radiating Pyongyang and South Korea’s Seoul). Finally, the China-Mongolia-Russia economic corridor will play the leading roles in Northeast Asia.

The areas along the China-Mongolia-Russia cross-border HSR are the important load bearing areas for the construction of China-Mongolia-Russia economic corridor. Although the grand blueprint for the construction of China-Mongolia-Russia HSR is ambitious, the potential risks of China-Mongolia-Russia HSR construction could not be ignored. First, the China-Mongolia-Russia HSR crosses the central plateau and southern Gobi desert of Mongolia. There are natural risks such as frozen soil disaster, overgrazing, grassland degradation, desertification and other natural disasters along the China-Mongolia-Russia economic corridor. Second, Mongolia, Russia’s Zabaikalsk Territory and Republic of Buryatia have the economic risks such as the backward infrastructure level, sparse population number and weak economic development level. Third, China’s Heilongjiang Province, Inner Mongolia Autonomous Region and Jilin Province have the social risks such as the low level opening to the outside world. Mongolia has the social risks such as the high unemployment rate. Fourth, with the improvement of the development speed and the spatial polarization of economic activities, the emissions of SO_2_, NO, chemical oxygen demand and other pollutants have been increasing along the China-Mongolia-Russia economic corridor. China-Mongolia-Russia economic corridor has the ecological environment risks such as the reduced biodiversity, degraded ecosystems, sensitive and fragile ecological environment. In the future, avoiding and solving the natural, economic, social and ecological environment risks will become the top priority of China-Mongolia-Russia HSR construction and the sustainable development of China-Mongolia-Russia economic corridor.

## 6. Conclusions

Under the background of “the Belt and Road” and “China-Mongolia-Russia economic corridor” initiatives, the urban accessibility level, regional accessibility pattern, and regional spatial effect were studied along China-Mongolia-Russia HSR west and east lines. The conclusions are as the following.

(1) The operation of China-Mongolia-Russia HSR will greatly improve the urban accessibility level, which will shorten the whole journey time to two days along China-Mongolia-Russia economic corridor. The regional space-time convergence effect will be very strong in the China-Mongolia-Russia HSR environment. Before and after the opening of China-Mongolia-Russia HSR, the urban accessibility level and urban accessibility improvement degree of the east line are stronger than those of the west line. The accessibility level of different countries differs: China > Russia > Mongolia. The accessibility improvement degree of different countries also differs: Mongolia > Russia > China.

(2) Spatially, before and after the opening of China-Mongolia-Russia HSR, the accessibility superior elements show a decreasing trend from Chinese cities via the China-Mongolia border and China-Russia border, to Russia. The accessibility improvement degree of the cities, which are located in the middle of the line is stronger than those cities at the beginning and end of the line. The superior areas of accessibility improvement will be concentrated in the series areas of Ulan Ude-Irkutsk-Krasnoyarsk-Novosibirsk-Omsk, the Erenhot-Zamyn Uud (border ports of China and Mongolia) and Manzhouli-Zabaykalsk (border ports of China and Russia).

(3) Affected by the China-Mongolia-Russia HSR environment, the gradual change and remodeling of regional spatial pattern will become more and more complicated along the China-Mongolia-Russia economic corridor. The spatial polarization effect of China-Mongolia-Russia HSR axial belt will be further enhanced, and the regional population and economic gravity centers will move to the China-Mongolia-Russia HSR axial belt. The internal boundary effect of the China-Mongolia-Russia HSR axial belt will disappear, and the integration effect of boundary areas and boundary cities will appear. New HSR economic growth poles will occur, promoting the formation of point-axis system of the China-Mongolia-Russia economic corridor. The regional trade creation and transfer effects will become obvious, and the cross-border investment scale will be deepened.

## Figures and Tables

**Figure 1 ijerph-19-10266-f001:**
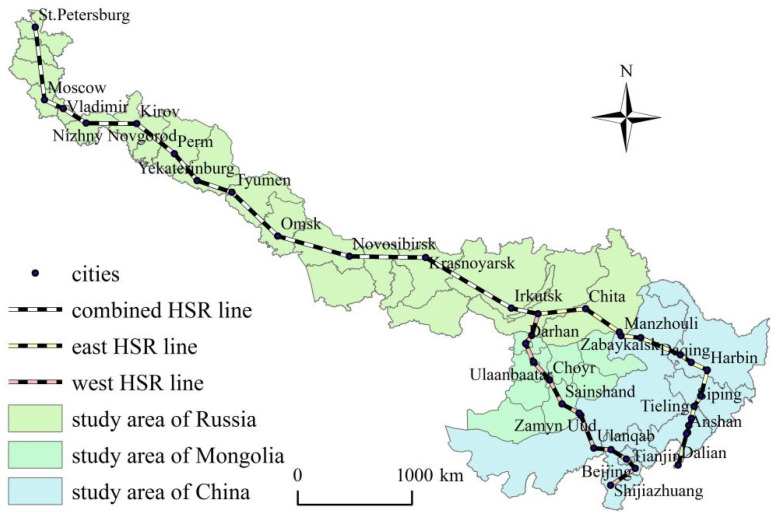
Sketch map of the study area.

**Figure 2 ijerph-19-10266-f002:**
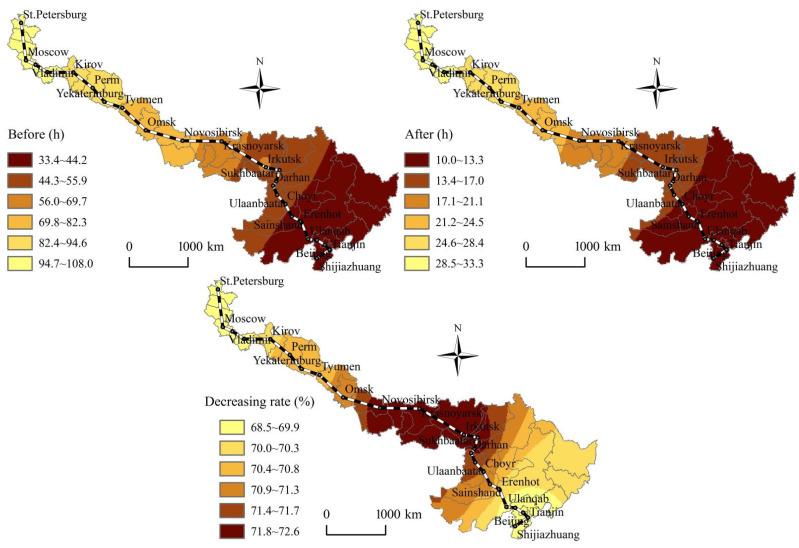
Spatial pattern changes of the weighted average travel time before and after the China-Mongolia-Russia HSR west line opening.

**Figure 3 ijerph-19-10266-f003:**
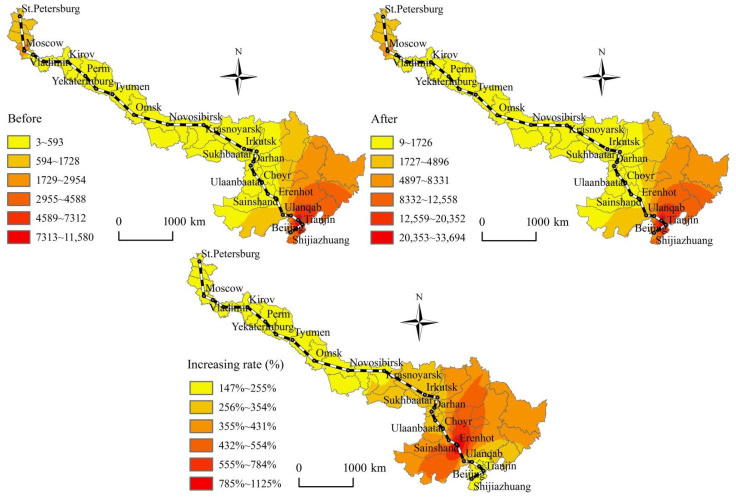
Spatial pattern changes of the economic potential before and after the China-Mongolia-Russia HSR west line opening.

**Figure 4 ijerph-19-10266-f004:**
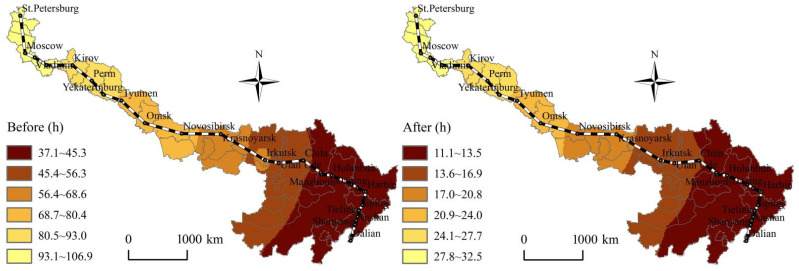
Spatial pattern changes of the weighted average travel time before and after the China-Mongolia-Russia HSR east line opening.

**Figure 5 ijerph-19-10266-f005:**
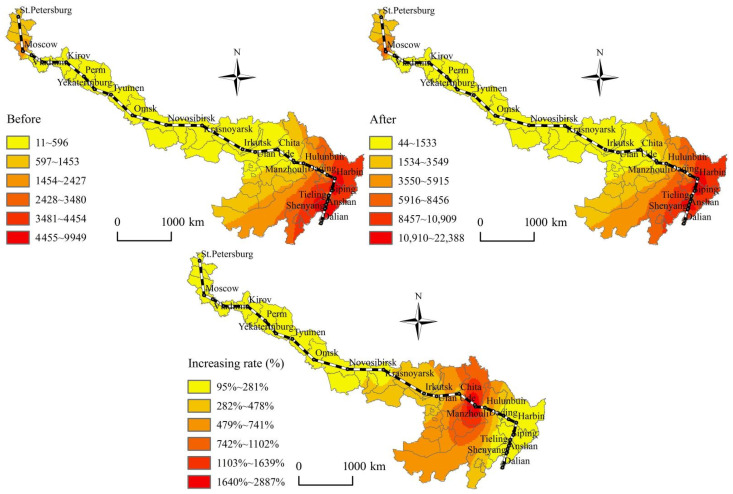
Spatial pattern changes of the economic potential before and after the China-Mongolia-Russia HSR east line opening.

**Table 1 ijerph-19-10266-t001:** Data sources.

Indicator		Data Sources
Transportation	Inter-city railway operation length	Reference [5]
	Intercity operation time of CST	China-Mongolia-Russia international train timetable
	Station retention time of CST	China-Mongolia-Russia international train timetable
	Intercity operation time of HSR	According to reference [36], *t = L/V*, *L* is the inter-city railway operation length, and *V* is the HSR environment design speed (HSR environment refers to the railway system that can provide trains with the highest speed above 200 km/h, and *V* is 200 km/h)
Population and economy	population and economic indicators of Chinese cities	«China Urban Statistical Yearbook 2021»
	population and economic indicators of Mongolian cities	the official website of the Mongolian Bureau of Statistics “http://www.nso.mn/”(accessed on 24 December 2019)
	population and economic indicators of Russian cities	the official website of the Russian Bureau of Statistics “https://www.gks.ru”(accessed on 24 December 2019)
Exchange rate	Mongolia’s tugrik is converted into Chinese RMB	the monthly average foreign exchange rate of the Bank of Mongolia from January to December 2020
	Russia’s ruble is converted into Chinese RMB	the cumulative average of the central parity of ruble exchange rate in the annual report of China foreign exchange administration from January to December 2020

**Table 2 ijerph-19-10266-t002:** Weighted average travel time and economic potential of cities along the China-Mongolia-Russia west line.

Countries	Cities	Weighted Average Travel Time (h)			Economic Potential			Countries	Cities	Weighted Average Travel Time (h)			Economic Potential		
		before	after	Decreasing Rate	before	after	Increasing Rate			before	after	Decreasing Rate	before	after	Increasing Rate
Russia	St.Petersburg	108.0	33.3	69.2%	615	1852	201%	Mongolia	Sukhbaatar	51.6	14.5	72.0%	2	7	269%
	Moscow	100.5	30.4	69.7%	2282	6751	196%		Darhan	49.8	14.2	71.5%	8	35	354%
	Vladimir	98.7	29.8	69.8%	51	140	176%		Ulaanbaatar	47.5	13.5	71.5%	240	893	273%
	Nizhny Novgorod	96.1	29.1	69.8%	176	518	194%		Choyr	45.1	13.0	71.2%	1	3	283%
	Kirov	92.7	27.5	70.3%	52	146	184%		Sainshand	43.4	12.4	71.3%	2	8	289%
	Perm	88.4	26.1	70.5%	119	349	192%		Zamyn Uud	41.3	11.9	71.2%	1	17	1113%
	Yekaterinburg	84.9	25.0	70.6%	200	589	195%	China	Erenhot	38.6	11.9	69.3%	14	178	1129%
	Tyumen	82.0	24.1	70.6%	96	288	199%		Ulanqab	36.6	11.1	69.6%	569	2026	256%
	Omsk	78.2	22.5	71.2%	106	298	182%		Zhangjiakou	34.4	10.5	69.5%	1320	4464	238%
	Novosibirsk	74.2	20.9	71.9%	143	421	194%		Beijing	33.4	10.0	70.0%	11,608	33,780	191%
	Krasnoyarsk	68.0	18.9	72.2%	88	288	228%		Tianjin	33.9	10.5	69.1%	5771	14,191	146%
	Irkutsk	59.3	16.2	72.6%	65	238	264%		Shijiazhuang	36.5	11.5	68.5%	2597	6953	168%
	Ulan Ude	55.1	15.1	72.5%	13	54	329%		average	63.1	18.6	70.6%	1046	2980	298%

**Table 3 ijerph-19-10266-t003:** Weighted average travel time and economic potential of cities along the China-Mongolia-Russia east line.

Countries	Cities	Weighted Average Travel Time (h)			Economic Potential			Countries	Cities	Weighted Average Travel Time (h)			Economic Potential		
		before	after	Decreasing Rate	before	after	Increasing Rate			before	after	Decreasing Rate	before	after	Increasing Rate
Russia	St. Petersburg	106.9	32.5	69.6%	619	1887	205%	China	Manzhouli	41.6	12.7	69.5%	70	1254	1693%
	Moscow	99.5	29.7	70.1%	2278	6824	200%		Hulunbuir	40.9	12.3	69.9%	797	2577	223%
	Vladimir	97.6	29.1	70.1%	50	141	184%		Qiqihar	37.9	11.7	69.1%	1987	5089	156%
	Nizhny Novgorod	95.3	28.4	70.2%	180	524	191%		Daqing	37.5	11.5	69.4%	1736	4218	143%
	Kirov	90.3	26.9	70.2%	47	148	213%		Harbin	37.2	11.1	70.1%	5759	14,486	152%
	Perm	85.8	25.5	70.3%	114	353	209%		Changchun	37.1	11.2	69.8%	5364	13,707	156%
	Yekaterinburg	82.3	24.5	70.3%	196	595	204%		Siping	37.4	11.4	69.6%	2203	5491	149%
	Tyumen	79.9	23.6	70.5%	99	291	194%		Tieling	37.7	11.6	69.3%	2300	5687	147%
	Omsk	76.2	22.1	71.0%	106	301	185%		Shenyang	38.1	11.7	69.4%	10805	24,397	126%
	Novosibirsk	71.6	20.5	71.4%	137	425	211%		Liaoyang	38.6	11.9	69.2%	2730	5573	104%
	Krasnoyarsk	65.9	18.7	71.7%	85	287	236%		Anshan	38.9	12.0	69.0%	5428	10,341	91%
	Irkutsk	57.9	16.1	72.2%	58	219	277%		Dalian	42.2	12.8	69.7%	2903	8036	177%
	Ulan Ude	53.9	15.1	72.1%	10	42	304%		average	60.1	17.8	70.3%	1707	4189	351%
	Chita	49.2	13.8	72.0%	33	150	351%								
	Zabaykalsk	44.3	12.7	71.3%	2	62	2988%								

## Data Availability

The data used to support the findings of this study are available from the corresponding author upon reasonable request.

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
