# Peer review of "Cross-Border Accessibility and Spatial Effects of China-Mongolia-Russia Economic Corridor under the Background of High-Speed Rail Environment"

_ijerph, 2022, doi:10.3390/ijerph191610266_

Round 1

Reviewer 1 Report

This paper takes the western and eastern railways of the China-Mongolia-Russia economic corridor as the research object. In the circumstance of high-speed rail (HSR) environment, the dynamic change of the accessibility of cities is measured by the accessibility indicators before and after the operation of China-Mongolia-Russia HSR.  And the regional spatial effect affected by the China-Mongolia-Russia HSR accessibility is analyzed. From the perspective of topic selection, the research object of this paper contains of three countries of China, Mongolia and Russia. At this stage, there is less research on the whole region of China-Mongolia-Russia economic corridor, and the selection of China-Mongolia-Russia HSR has some new ideas. At the same time, this study plays a theoretical guiding role in the implementation of the Northeast Revitalization Strategy, the expansion of bilateral cooperation, and the construction of the China-Mongolia-Russia HSR. It has some important research value based on the theoretical front. However, the content of spatial effect analysis is a little weak. It is suggested to analyze the spatial effect of China-Mongolia-Russia economic corridor from the perspective of cross-border trade creation and cross-border trade transfer.

Reviewer 2 Report

The current paper has examined the Cross-border accessibility and spatial effects of China-Mongolia-Russia economic corridor under the context of high-speed rail environment. The subject investigated has some worth, and the paper may be accepted for publishing following major revisions. Authors are invited to carefully address the concerns/comments below.

1.      The abstract is very lengthy and confusing; consider reducing it significantly and also organize it properly.

2.      At the start of the introduction, authors should provide a brief overview of  HSR, its importance, and its benefits, followed by Belt and Road initiative as provided.

3.      The study of related literature is deficient, and should be substantially improved.  

4.      Further study motivations are weak. At the end of the introduction, it is recommended that specific objectives and contributions of the current study be provided.

5.      Why were only two indicators (travel time and economic potential) selected for analysis? Why not other important indicators (such as comfort, safety, etc.)not considered?

6.       Section 2.2., research methods should be further elaborated to facilitate understanding of a wider audience.

7.      It will be more appropriate to add a summary table for “data sources” used.

8.      Conclusions are again very long. It should be made precise and to the point summarizing the key points/findings.

9.      Study limitations and outlook for forthcoming studies should be included/refined.  

10.   The discussion section should contain a critical analysis of the study results in light of similar previous studies, not duplications of results already presented.

11.   The paper should be thoroughly checked for English and grammar editing as the current text has several typos and mistakes. 

Round 2

Reviewer 2 Report

The authors have provided satisfactory responses to most of my earlier comments. However, it is recommended to re-consider the following minor concerns.

Point#5: I still believe that considering additional indicators as suggested previously will add value to this paper. 

Point#10: The discussion section has not been improved as advised. statting "compared with earlier studies [16-28]....." in lumps and then putting results as presented is not justified. I expect the authors to include critical discussion and comparison with specific relevant studies. Like what they have achieved and whether the current study findings support or do not support their findings and potential justification for deviation. 

Point# 11: there are still a few typos and grammar issues and the manuscript needs a careful review for language editing. 

New Comment:

It is suggested to improve the formatting and resolution of Figures 1-5. Also, Figure 2 should be brought to the page center like others. 
